# Tomato Yellow Leaf Curl Virus (TYLCV) Promotes Plant Tolerance to Drought

**DOI:** 10.3390/cells10112875

**Published:** 2021-10-25

**Authors:** Moshik Shteinberg, Ritesh Mishra, Ghandi Anfoka, Miassar Altaleb, Yariv Brotman, Menachem Moshelion, Rena Gorovits, Henryk Czosnek

**Affiliations:** 1Robert H. Smith Faculty of Agriculture, Food and Environment, Institute of Plant Sciences and Genetics in Agriculture, The Hebrew University of Jerusalem, Rehovot 7610001, Israel; moshik696@gmail.com (M.S.); ritesh.mishra@mail.huji.ac.il (R.M.); menachem.moshelion@mail.huji.ac.il (M.M.); rena.gorovits@mail.huji.ac.il (R.G.); 2Faculty of Agricultural Technology, Al-Balqa Applied University, Al-Salt 19117, Jordan; anfoka@bau.edu.jo (G.A.); miassaraltaleb@yahoo.com (M.A.); 3Faculty of Life Sciences, Ben Gurion University of the Negev, POB 653, Beer-Sheva 8410501, Israel; brotmany@post.bgu.ac.il

**Keywords:** begomovirus, drought, plant-virus interaction, osmo-protective metabolites

## Abstract

A growing body of research points to a positive interplay between viruses and plants. Tomato yellow curl virus (TYLCV) is able to protect tomato host plants against extreme drought. To envisage the use of virus protective capacity in agriculture, TYLCV-resistant tomato lines have to be infected first with the virus before planting. Such virus-resistant tomato plants contain virus amounts that do not cause disease symptoms, growth inhibition, or yield loss, but are sufficient to modify the metabolism of the plant, resulting in improved tolerance to drought. This phenomenon is based on the TYLCV-dependent stabilization of amounts of key osmoprotectants induced by drought (soluble sugars, amino acids, and proteins). Although in infected TYLCV-susceptible tomatoes, stress markers also show an enhanced stability, in infected TYLCV-resistant plants, water balance and osmolyte homeostasis reach particularly high levels. These tomato plants survive long periods of time during water withholding. However, after recovery to normal irrigation, they produce fruits which are not exposed to drought, similarly to the control plants. Using these features, it might be possible to cultivate TYLCV-resistant plants during seasons characterized by water scarcity.

## 1. Introduction

In regions of extreme drought and heat, such as the Middle East, cultivating tomato is a real challenge. In addition, tomato yellow leaf curl virus (TYLCV), a whitefly-transmitted begomovirus, thrives in the same regions. Often, the combination of these biotic (TYLCV) and abiotic (heat, drought) stresses cause a yield reduction up to total loss. Therefore, there is a need to improve the tolerance of crops to the combined effects of high temperature, drought, and insect-transmitted diseases. By understanding the complex mechanisms of plant responses to these stresses, it might be possible to develop strategies relying on genes, proteins, and metabolites expression profiling, aimed at adapting these plants to harsh environments. This is not unfeasible since we have previously shown that TYLCV infection could prompt protection from heat stress [1].

Begomoviruses, the TYLCV family among them, are transmitted in nature by the whitefly *Bemisia tabaci* and infect a large number of agricultural crops. TYLCV possess a single genomic molecule encapsidated in a 20 × 30 nm geminate particle. The circular single-stranded DNA genomic molecule of about 2800 nucleotides encodes two genes, V1—or coat protein CP, and V2—or movement protein MP; the genome complementary stains encodes four genes, C1—or replication initiator protein Rep, C2—a transcription activating factor, C3—a replication enhancer, and C4—a RNA suppressor silencing contributing to virus spread [2].

To replicate, TYLCV reprograms the plant cell cycle into the S phase and interacts with host factors to create a permissive environment [3]. To ensure a successful infection, TYLCV restrains its destructive effects in infected tomatoes. It does not induce a hypersensitive response (HR) and cell death (CD). On the contrary, TYLCV suppressed CD induced by other factors by downregulating the genes Hsp90 and Sgt1 (a HSP90 co-chaperone) [4], which otherwise lead to the accumulation of damaged ubiquitinated proteins, and by downregulating the ubiquitin 26S proteasome (UPS) degradation mechanism [5].

TYLCV is able to manipulate the plant heat stress response and moderate the effects caused by heat on the plant [1]. It was proposed that one of the potential mechanisms of suppression of heat stress response was based on the ability of all six viral proteins to interact with the main heat shock transcription factor HSFA2. In tomato cell cultures subjected to heat stress, HSFA2 accumulated to high levels during prolonged exposition to heat [6]. HSFA2 regulated the expression of members of the ascorbate peroxidase (APX) family, which are major scavengers of intercellular H_2_O_2_ and prevented ROS overproduction [7]. Capturing HSFA2 by viral proteins inhibited its translocation into nuclei and, consequently, downregulated the transcriptional activation of heat stress response genes [1]. In TYLCV-infected tomatoes exposed to drought, the activation of HSFA2 was reduced, and two other key stress transcription factors, HSFA1 and HSFB1, were also mitigated. However, a major effect of TYLCV was the reallocation of some dominant carbohydrates and amino acids from shoots to roots, which prepared them to the effects of drought [8].

Therefore, by suppressing an acute stress response leading to cell death, TYLCV created a proper environment for its successful replication and spread. In the current study, we used the capacity of TYLCV to protect tomatoes against drought, not only in TYLCV-susceptible, but mainly in TYLCV-resistant tomato germplasm. TYLCV-resistant plants, containing virus amounts that did not prevent their development and fruit production, demonstrated enhanced tolerance to drought. After prolonged growth under severe water deficit, tomatoes were able to recover and yield.

## 2. Materials and Methods

### 2.1. Viruses, Insects and Plants

TYLCV from Jordan [9] and from Israel [10] were used in location to inoculate tomato plants (these two virus isolates are stains from the same virus since their sequences show more than 95% identity). Whiteflies (*Bemisia tabaci* B biotype, termed also MEAM1) were used to inoculate tomato plants. Seedlings (3 weeks after sowing) were placed in 50-mesh net cages with adult viruliferous whiteflies for 7 days (dpi) (approximately 50 whiteflies per plant). Plants from the TYLCV-susceptible 967 and the TYLCV resistance line GF967 were used throughout the current study. Line 967 (denominated here S-967) was obtained from the Jordanian Ministry of Agriculture [11]. Line GF967 (denominated here R-GF967) was developed in Jordan; it resulted from a cross between line 967 and a TYLCV-resistant line previously bred in Guatemala [12], based on the resistant germplasm selected in Israel [13], which we used in our previous studies [14,15].

### 2.2. Drought Treatment

In the Jordan Valley, at the Research Station of the Al Balqa Applied University, Jordan, tomatoes were grown in plastic pots filled with a mixture of peat moss and perlite. Plants were kept in an insect-proof greenhouse at temperatures of 25–30 °C. Three weeks after TYLCV infection, plants were divided into two groups. One group was irrigated as usual, while in the other group irrigation was totally suppressed. Symptoms associated with drought were monitored every 3 days during an 18-day-long period (0, 3, 6, 9, 12, 15, 18 dd). The experimental growth in the Jordan valley was repeated three times, each consisting in four groups of plants contained 15 seedlings.

In a climatized experimental greenhouse on the grounds of the Faculty of Agriculture at Rehovot, Israel, tomato seedlings of S-967 (S) and R-GF967 (R), TYLCV-infected (V) and not infected (0), were grown in ambient conditions in four-liter pots (1 seedling in each pot) with peat-based potting soil. Both were ferti-irrigated to the daily pot capacity with tap water (W) or under complete water withholding (D). In a specially conceived research greenhouse located at the Faculty of Agriculture, coined as iCORE (https://plantscience.agri.huji.ac.il/icore-center, last accessed 24 October 2021), plants were grown at 31–15 °C, with humidity of 30–80%, under natural light conditions. Drought-exposed tomatoes were grown without watering during 14 dd. The well-irrigated and drought-exposed tomatoes were followed continuously and automatically all along the experiment. The experiment in the iCORE greenhouse was repeated three times, and each group of plants contained 10 seedlings. At the end of the iCORE experiment, dry weights of shoots and roots (g) were measured. Shoots were removed from the plants and dried in a hot air oven at 60 °C, until no further reduction in weight was measured. Roots were washed thoroughly to remove the soil, then dried similarly to shoots.

### 2.3. Analysis of Metabolites

Tomato leaf samples were collected from S-967 and R-GF967, infected or not with TYLCV, during the 18 days of the drought treatment in Jordan (0, 3, 6, 9, 12, 15 and 18 dd). The leaf samples were analyzed for metabolite patterns at the Planck-Institute in Golm, Germany. Metabolic profiles covering soluble sugars and amino acids were analyzed using a gas chromatograph Leco Pegasus time-of-flight mass spectrometer (GC-MS, Leco). Leaf material (10 mg dry weight, 3 replicates per treatment and line) was ground, and metabolites were extracted and derivatizated [16]. The acquisition parameters were previously described in detail [17]; peak detection, retention time alignment, and library searching were performed with the “TargetSearch” package [18]. The amount of amino acids or sugars in the leaves of plants regularly watered at the first sampling time (0 dd) was considered as 100. Levels of leaf amino acids and sugars in all plants, used throughout experiments, were estimated relative to this value. Metabolome data were analyzed by principle component analysis (PCA) and partial least-squares discriminant analysis (PLSDA) using the statistics software MetaboAnalyst 5.0) (freely available at www.metaboanalyst.ca, accessed on 24 October 2021; https://pubmed.ncbi.nlm.nih.gov/34019663/, accessed on 24 October 2021).

### 2.4. Measurements of Quantitative Physiological Traits in iCORE Greenhouse

Fertigation treatments of tomato growth in the greenhouse were controlled by the Plantarray system (PlantDitech Ltd., Yavne, Israel). In addition, the functional Plantarray system was used to monitor the whole plants’ physiological behavior during the 14-day period by controlling the quantity of water used for irrigation. Each Plantarray unit had a personalized controller, collecting the irrigation conditions as well as the data collection of each plant separately grown in a pot. The data were analyzed by SPAC-Analytics (Plant-Ditech, https://www.spacanalytics.com/, last accessed 24 October 2021), monitored online by web-based software. Quantitative physiological traits of the plants were determined as previously reported [19], and equations were implemented in the SPAC analytics software: daily transpiration (grams per day, g), E-normalized transpiration rate (g water/g plant/min), transpiration rate (g/min) vs. soil water content, and water use efficiency (WUE) (g/g). Calculated plant weight (CPW) (g) was determined as the sum of the initial plant weight (seedling weight measured manually on the first day of the experiment) with the cumulative transpiration multiplied by the WUE, identified as the ratio between the gain of daily weight and the daily transpiration. 

### 2.5. Protein Immunodetection

Exactly weighted 20-mg dried leaves were drill-homogenized in 300 µL of standard SDS-PAGE loading buffer supplemented with 2% SDS. Samples were boiled for 10 min and centrifuged for 10 min at 10,000× *g*; 30 µL of supernatants were subjected to SDS-PAGE. Western blotting and immunodetections were performed as described before [15]. Antibodies against the following proteins were purchased from Agrisera (Sweden): HSP70 (AS08371-100), HSP90 (AS08346), and BiP (AS09481). Incubation with primary antibodies was followed by exposure to a secondary goat peroxidase coupled antibodies (Agrisera, Sweden). ECL detection was performed (Amersham, UK). Each immunodetection was repeated at least three times for each set of plants and tissues. Blots were documented using an Image Quant LAS500 imager (GE Healthcare Life Sciences, Piscataway, NJ, USA). Only the relevant parts of the blots are shown in the figures.

### 2.6. Drought Recovery Treatment

In a climatized experimental greenhouse in Rehovot, seedlings of R-GF967 plants (three weeks after sowing) were grown in four-liter pots, one seedling per pot. Three weeks after planting, half the plantlets were caged with viruliferous whiteflies for 7 days (approximately 50 whiteflies per plant) before removal of the insects with imidacloprid. After three days, the two groups of plants were devised into two groups: one was watered normally (HW and VW) while the other was subjected to complete water withdrawal (HD and VD) for the next 18 days. Then, normal irrigation was resumed for all four groups of plants. After two months, all the fruits were collected and weighted. Five plants from each group were randomly chosen for fruit collection and weight. The experiment was repeated three times.

### 2.7. Statistical Analyzes

Three independent biological replicates were used in each experiment. The levels of amino acids and sugars were determined using three technical replicates. Comparisons were performed between watered and drought-treated plants for each time point. Each tissue and time point was considered as an independent experiment; therefore, a one-way analysis of variance. Significant differences were analyzed by the Student’s *t*-test using the statistical analysis software package JMP Pro Version 10.0 (SAS Inc., Cary, NC, USA, free access at https://www.jmp.com/ensues/software/data-analysis-software.html, accessed on 24 October 2021). Error bars represent standard errors obtained using an Excel spreadsheet. Statistical analyses were also performed based on partial least-squares discriminant analysis (PLSDA).

## 3. Results

### 3.1. TYLCV Infection of a TYLCV-Resistant Tomato Line Enhanced Plant Survival When Grown under Water Deficit

During July–August 2020, in the Jordan Valley, TYLCV-susceptible (S-967) and TYLCV-resistant (R-GF967) tomato lines were grown in one-liter pots filled with peat moss and perlite and kept in an insect-proof greenhouse. The seedlings were either TYLCV-inoculated using viruliferous whiteflies during 7 days (dpi) or left uninfected. Afterwards, 30 plants (15 TYLCV-inoculated and 15 not) of each line were subjected to drought by withholding irrigation for the next 18 days (drought days, dd). In this case, water was not supplied, the only source of water was the morning dew. Drought-induced wilting was recorded 0, 3, 6, 9, 12, 15, and 18 dd after treatment. Three independent rounds of growth were performed. During the 18-day-long experiment, leaf samples were collected at the days recorded and freeze-dried for further analyses of osmolytic sugars and amino acids, and stress-response proteins (Figure 1 and Figure 2). Uninfected S-967 plants (virus-susceptible, not infected, drought, —or S0D) started to show mild wilting at 3–6 dd; from 9 dd, leaves turned yellow and dry, and some of the tomatoes died. By comparison, S-967-infected tomatoes (virus-susceptible, infected, drought, —or SVD) survived better and grew for a longer time, showing mild wilting and leaf yellowing practically until the end of the experiment. Uninfected R-GF967 (resistant, not infected, drought, —or R0D) tomatoes demonstrated a higher extent of survival than S0D. The infected R-GF967 plants (resistant, infected, drought, —or RVD) showed the best survival rate under severe drought conditions. Even after 18 dd, none of the RVD tomato plants had collapsed, leaves of only five plants turned yellow; the other 10 presented only mild wilting. Three independent biological replicates were performed with similar effects. These results demonstrate that TYLCV infection increased the tolerance of tomato plants to drought in TYLCV-susceptible, but most remarkably, in TYLCV-resistant germplasms.

### 3.2. Osmo-Protectants (Sugars, Amino Acids) Accumulate in TYLCV-Infected Tomatoes Grown in Conditions of Water Deficit

To protect their cells from osmotic stress and to sustain turgor, plants enhance the concentration of osmo-protectants such as amino acids, proteins, and sugars [20,21]. In different plants exposed to water stress, the accumulation of soluble sugars might contribute to osmoregulation [22].

The amounts of sucrose, glucose, and fructose were analyzed by GC-MS in leaf samples, collected from S0D, SVD, R0D, and RVD plants along the 18 dd-long experiment in Jordan. Increased amounts of sugars were observed in all experimental plants (Figure 1A). The levels of sucrose, glucose, and fructose in S-967- and R-GF967-uninfected leaves were comparable, while TYLCV-infected S-967 tomatoes (SVD) showed a mitigation of sugars increase very early during the water withholding growth. In leaves of TYLCV-infected R-GF967 (RVD), such suppression was not observed, and sugars levels were similar to those of R0D, or even slightly higher.

The levels of several amino acids, known to be involved in response to drought [23,24], were analyzed by GC-MS, and compared between leaves of S-967 and R-GF967 tomatoes exposed to drought during the 18 days-long experiment. The amounts of proline, HO-proline, tryptophan, GABA, glutamine, valine, asparagine, and methionine were compared in S0D and SVD vs. R0D and RVD (Figure 1B). These selected amino acids were induced by drought to varying degrees and at different periods during exposure to drought. For example, the amounts of proline and of its derivative HO-proline increased during the entire 18-dd growth period. However, in S0D plants, this increase was milder than in R0D tomatoes. A surprising result was obtained in uninfected R-GF967 tomatoes, where levels of proline and HO-proline exceeded that in non-infected S-967 (R0D vs. S0D). The quantities of γ-aminobutyric acid (GABA), another osmolyte protecting cells against stresses [25], increased upon drought but only after 12 dd (Figure 1B). Alike-proline and HO-proline GABA amounts were higher, not only in RVD vs. SVD, but also in R0D vs. S0D. Besides GABA, three other members of glutamate family, glutamate, glutamine, and arginine, were analyzed. Glutamic acid was not induced by drought (not shown), similar to the absence of drought-dependent induction of this amino acid in the tomato commercial cultivar Ikram [8], contradicting data in the literature [26]. Stress-induced levels of glutamine and arginine were higher in R-GF967 than in S-967 (shown for glutamine in Figure 1B). Tryptophan, asparagine, methionine, and valine were induced by water withholding, especially in leaves of R-GF967. The current analysis did not reveal any effect of water deficit on the levels of alanine, threonine, leucine, and iso-leucine in S-967 and R-GF967 tomatoes during the 18-dd submission to drought (not shown). The levels of osmolytic amino acids induction in R0D were higher than in S0D, especially after 12 dd (Figure 1B). In most SVD plants, TYLCV caused an alleviation of drought-dependent increase in amino acids. In RVD tomatoes, this intensification was kept approximately at the same level during prolonged exposure to stress (15 and 18 dd).

The amounts of the metabolites, including soluble sugars and amino acids, obtained for each tomato genotype, infected or not, during the 18 days of the experiment were statistically analyzed by partial least-squares discriminant analysis (PLSDA) and principle component analysis (PCA) which showed a clear discrimination between tomato genotypes (Figure 1C). All the data for a given genotype are represented by a colored circle. Red: R0D; green: RVD; light blue: S0D; dark blue: SVD. Results show that R-GF967, infected or not, cluster together; on the contrary, S-967, infected and uninfected, behave differently, and more differently than R-GF967 plants.

The results showed that TYCV-resistant and TYLCV-susceptible genotypes behaved differently upon drought. In resistant and infected plants, stable patterns of osmoprotectants were maintained along the entire growth under stress. On the contrary, in susceptible plants, infected and uninfected plants behaved differently, characterized by virus-induced mitigation.

### 3.3. Enhanced Stabilization of HSPs Patterns in Leaves of Tomatoes Exposed to Prolonged Drought

Any stress, by definition, involves protein denaturation, which in turn requires the recruitment of heat shock proteins to limit damages (HSPs). HSPs are not only involved in the heat stress response, but also in response to various environmental stresses, including drought, oxidation, osmosis, salt, UV, and infection by different pathogens [27]. The passage from sufficient to inadequate water amounts activate a cytoplasmic response in which heat shock transcription factors are activated.

The patterns of the major members of the HSP family—HSP70, BiP, and HSP90—were followed in the leaves of uninfected and TYLCV-infected tomato lines, i.e., S-967 and R-GF967. Withholding of water from S-967 and R-GF967 tomatoes did not cause a significant increase in the abundances of HSPs (Figure 2). Some increases in the amounts of HSP70 and HSP90 were observed, while patterns of BiP did not change. The amounts of HSPs in uninfected S0D and R0D plants were comparable. TYLCV infection caused improved stability of all HSPs in SVD and RVD. In RVD, the improvement was somehow more apparent for BiP and HSP90 (Figure 2). These results pointed to an enhanced stabilization of HSPs patterns in leaf samples of infected tomatoes exposed to prolonged drought.

### 3.4. Transpiration, Whole Canopy Stomatal Conductance and Physiological Drought in TYLVC-Susceptible and Resistant Tomatoes Grown under the Water Deficit

The influence of virus infection on the whole-plant physiological parameters and water balance was previously studied in uninfected and infected TYLCV-susceptible tomato cultivar Ikram [8], using the high-tech Pantarray-functional-phenotyping platform located in the iCORE greenhouse (https://plantscience.agri.huji.ac.il/icore-center, accessed on 24 October 2021), specially designed to extract simultaneously various individual information from numerous plants, under a variety of controlled conditions (e.g., water availability, nutrient level, biomass, stress index, etc.) [19]. In the current study, we compared quantitative physiological traits in watered TYLCV-resistant vs. TYLCV-susceptible tomato lines, infected or not (RVW, SVW; ROW, SOW), and subjected to drought or not (RVD, SVD; ROD, SOD), as tested previously in Jordan. Tomatoes were grown during 14 dd in the iCORE greenhouse settings and stress-related parameters were recorded individually during the 14 dd-long experiment.

TYLCV infection in watered TYLCV-susceptible tomatoes (SVW) resulted in the reduction of transpiration, transpiration rate per unit leaf area (E), and physiological drought point (Θ_crit_) levels [19]. In infected plants subjected to drought, these values already declined slower in infected plants, rather than in uninfected. Virus infection caused a rise in water balance stability, which may allow plants to adapt to drought.

The changes in daily transpiration of watered TYLCV-susceptible and TYLCV-resistant tomato plants (S0W and R0W) were comparable along the experimental growth period (Figure 3A). Under drought, these two lines (S0D and R0D) also behaved quite similarly, which was not entirely expected because of the differences in the osmo-protectant patterns described above (Figure 1). TYLCV infection led to a decrease in the transpiration in SVD, which could be explained by virus-dependent attenuation of S-967 growth. A slower decrease in transpiration rates was detected in RVWR vs. R0W, likely because the plant development was much less dependent on the presence of the virus. TYLCV attenuated the decrease in the transpiration of SVD and RVD tomatoes in comparison with non-infected plants, S0D and R0D (Figure 3A).

Transpiration rate per plant weight (E) was similar in S0W and R0W tomatoes. TYLCV infection led to their decline from the beginning of growth, in both SVW and RVW (Figure 3B). TYLCV infection delayed the decrease in E, prolonging the water balance for several days. If the decline of E in S0D and R0D plants started after 6 dd, in SVD and RVD, it only did so between 9 and 10 dd.

The physiological drought point (Θ_crit_), which reflected the midday transpiration rate vs. soil water content, was also overdue in infected vs. non-infected tomatoes (Figure 3C). Therefore, the virus helped shift Θ_crit_, when the soil water content started to limit the transpiration rate, promoting plant survival. SVD Θ_crit_ was estimated at 0.258 and S0D at 0.3; the difference between RVD Θ_crit_ (0.256) and R0D (0.308) was even higher. In TYLCV-infected plants, the water use efficiency (WUE) was higher (though not significantly) than in uninfected plants (Figure 3C), whether the plants were resistant or susceptible to TYLCV (12% RVW vs. R0W; 12% SVW vs. S0W.

The next physiological parameter measured was the gain of fresh weight (CPW) of the experimental plants, calculated during the entire growth period. During normal irrigation, S0W gained more fresh weight compared to SVW plants (Figure 3D). Comparison of R0W and RVW showed a smaller difference, as expected when comparing S0W and SVW. In drought conditions, CPW of SVD exceeded that of S0D by approximately 17%, while the CPW of RVD enhanced that of R0D by 12% (Figure 3D). The CPW of S0D reached a plateau at 10 dd, while that of SVD did it at 13 dd, a pattern similar to that obtained for R plants. Furthermore, the CPW of R0D reached a plateau at 10 dd, while that of RVD, reached between 12–13 dd. The CPW calculation suggests that, in plants exposed to drought, the gain of weight was due to virus infection, but not because of the identity of the treated tomato line.

### 3.5. Shift of Biomass from Shoot to Root Caused by the Presence of TYLCV

After 14 dd in the iCORE greenhouse, the shoots and roots of each plant from every group were dried and weighted. A decrease of 65–70% shoot dry weight was detected in HDS plants compared to a decrease of approximately 55% in HDR (Figure 4). Roots dry weight decreased approximately three times in HDS and two times in HDR. Virus infection affected the dry weight of VWS plants, decreasing the weight of shoots by 50%, compared to a decrease of 20–25% in roots. The influence of TYLCV on the dry weight of VWR vs. HWR was expected to be lower. VWR shoots lost only 15–17% weight and roots did not lose any biomass at all. After growth in water-deficit conditions, dry weight loss of around 28% was found in shoots and 25% in roots of VDS and VDR (Figure 4).

The root to shoot ratio (R/S) of TYLCV-infected S-967 was higher than of uninfected tomatoes already during growth under normal irrigation (Figure 4, HWS vs. VWS). During growth under drought, R/S ratio of VDS increased, while that of HDS remained unchanged. For R-GF967 plants, R/S ratio of VWR was higher than that of HWR. During growth under drought, both VDR and HDR stayed approximately the same; the principal difference in allocation of biomass from shoot to root was caused by the presence of TYLCV.

### 3.6. Infected TYLCV-Resistant Plants Subjected to Drought for 14 Days Were Able to Recover and Produce Fruits

After 14 dd, all R-GF967 tomatoes, subjected to drought or not (R0W, RVW, R0D, RVD), were returned to normal irrigation, allowing them to recover from the stress caused by drought. The TYLCV-susceptible S-967 plants were not included in the recovery experiments because TYLCV infection had irreversibly destroyed the plant [2]. During the two months of recovery, tomato fruits were collected. The fruits harvested from five individual plants chosen at random from each group ((R0W, RVW, R0D, RVD) were weighted. The experiment was repeated three times. R0W and RVW produced eight fruits, weighing altogether about 950 g (Figure 5A), which confirmed that the virus did not influence fruit setting in the TYLCV-resistant line. In comparison, exposure of uninfected plants to drought (R0D) also produced eight fruits but their weight decreased by about two thirds. However, RVD plants, treated as R0D, lost only 20–25% of fruit weight (Figure 5A). Altogether, when comparing the tomato fruits collected, it can be noticed that, although drought was associated with a smaller size of R0D tomatoes, it practically did not affect that of RVD tomatoes (Figure 5B). Hence, TYLCV-infected R-GF967 tomatoes successfully overcame an extended cessation of irrigation, and after a recovery period, produced tomato fruits as did untreated tomato plants.

## 4. Discussion

The ability of viruses to protect the infected plant hosts from negative effects of different environmental stresses is a topic that is becoming increasingly important in agricultural research. Undoubtedly, many viruses encountered in agricultural systems are pathogenic and affect yields; nonetheless, viruses may also co-exist with plants without harming them, and even benefit to them [28,29]. Some viruses are clearly beneficial to their infected host, protecting them from heat and/or water scarcity [1,30,31,32]. Increasing amounts of studies demonstrated the way viruses manage to increase the plant’s response to environmental stresses (reviewed in [33]). For example, four different RNA viruses, brome mosaic virus (BMV), cucumber mosaic virus (CMV), tobacco mosaic virus (TMV), and tobacco rattle virus (TRV), were shown to improve the infected plant’s tolerance to drought [30]. Examples of plant beneficial trade-off from infection with potato virus X (PVX) and plum pox virus (PPV) was described for *N. benthamiana* and Arabidopsis grown under drought conditions. Virus infection enhanced plant tolerance to drought by increasing salicylic acid amounts, in an abscisic acid (ABA)-independent manner [34]. Tobacco rattle virus (TRV) was shown to change the stress response of Arabidopsis to low temperature [35]; PVX—to increased environmental oxidation in *N. benthamiana* [36].

In a previous study [8], we demonstrated the ability of TYLCV infection to enhance the survival to drought of TYLCV-susceptible tomatoes. The main purpose of the current study was to compare the behavior of TYLCV-susceptible (S-967) with TYLCV-resistant (R-GF967) tomato lines grown upon water withholding. We predicted that the presence of the virus in the resistant R-GF967 tomatoes would cause a similar level of protection against drought, remain symptomless, and yield (contrary to the infected susceptible tomatoes S-967 who will decay upon drought). Indeed, viral infection of R-GF967 tomatoes caused an improvement in growth and survival. Drought did not lead to the collapse of RVD plants grown in the Jordan valley and in the greenhouse (Figure 1 and Figure 2). RVD plants showed mild wilting and leaf yellowing only after 18 dd. It must be mentioned that virus infection significantly improved drought tolerance of S-967 (SVD) tomatoes as well, which confirmed our previous results [8].

The capacity of plants to resist numerous environmental stresses is associated with the increase in different cellular osmoprotectants [37]. The levels of protective sugars and amino acids in tomato leaves improved upon drought in non-infected R-GF967 (R0D) compared to S-967 (S0D) (Figure 1A, B). Moreover, the patterns of most osmoprotectants analyzed, including HSPs (Figure 2), were more stable in R0D tomatoes, rather than in S0D tomatoes. Protein homeostasis was maintained longer in TYLCV-resistant tomatoes than in susceptible tomatoes exposed to stresses [14,38]. In the current study, in addition to HSPs, the well-balanced homeostasis of R0D plants was confirmed by the enhanced stability of such osmolytes as amino acids and soluble sugars (Figure 1).

TYLCV infection induces drought tolerance in TYLCV-susceptible tomatoes, S-967 line in this study, and in cv. Ikram [8]. Moreover, this study showed that the induction of drought tolerance is particularly pronounced in infected TYLCV-resistant plants (R-GF967), which accumulate less amounts of virus than susceptible plants, thus not affecting plant growth and development [15]. Our findings showed that stress-protective soluble sugars and amino acids in the leaves of uninfected, and especially in virus-infected R-GF967 tomatoes, resulted in the development of a buffering state, resulting in plant protection against drought during a prolonged time (Figure 1). Therefore, we propose that the establishment of stress osmo-protectant stability is the main feature underlaying increased drought tolerance of TYLCV-resistant tomatoes, which is utterly improved by virus infection.

Daily transpiration rates of S-967 and R-GF967 tomatoes were comparable (Figure 3). TYLCV infection attenuated transpiration in a similar way in both lines, stabilizing water balance in the infected tomatoes. The confirmation of the role of TYLCV in water balance was obtained by analyzing physiological parameters, such as E, Θ_crit_, and CPW (Figure 3). Θ_crit_ showed some benefits of RVD over SVD, while WUE was somehow better for SVW than for RVW. However, the effect of the virus on tomato water balance was independent of the origin of the tomato genotype since TYLCV infection of S-967 and of R-GF967 plants acquired the capacity to maintain a relatively stable water balance during prolonged drought.

Enhanced osmoprotectants in uninfected R-GF967 plants could be explained by a sufficiently less drought-dependent drop of shoot dry weight in R0D (around 55%) in comparison with S0D (65–70%) (Figure 4). R0D roots biomass was maintained even better under water stress. After TYLCV infection, the percentage of dry weight loss in shoots and roots was similar for S-967 and R-GF967, which could be explained by the dominant virus influence on plant survival under drought stress. The distribution of plant biomass between root and shoot, coined as R/S ratio, is an important criteria of plant capacity to adapt to environmental stresses. The gained R/S ratio points reflect an increased plant tolerance to drought stress [39]. The enhancement of R/S ratio was observed already in infected watered tomatoes and became more effective in drought-exposed plants (Figure 4). That increase in R/S ratio depended on the presence of TYLCV, but not on the nature of the tomato lines.

The main goal of farmers all around the world is to obtain high yields. In the current study, we showed that the use of TYLCV-infected and TYLCV-resistant tomatoes could overcome relatively long (at least 18 dd) periods of extreme drought, survive, and produce fruits after recovery (Figure 5). We have explored a concept based on the two apparent contradictions: on the one hand, extreme drought ultimately causes plant death; and on the other hand, TYLCV mitigates tomato disease to create a proper environment for its successful replication, allowing the plant to grow without irreversible damage.

Cultivating infected R-GF967 (or resistant cultivars similar to GF967) in conditions of water deficit is advantageous; TYLCV protects them from stress and does not interfere with their normal development. Thus, taking advantage of some of the virus effects, one can envisage cultivating TYLCV-infected and TYLCV-resistant plants in countries where drought is an acute problem.

## Figures and Tables

**Figure 1 cells-10-02875-f001:**
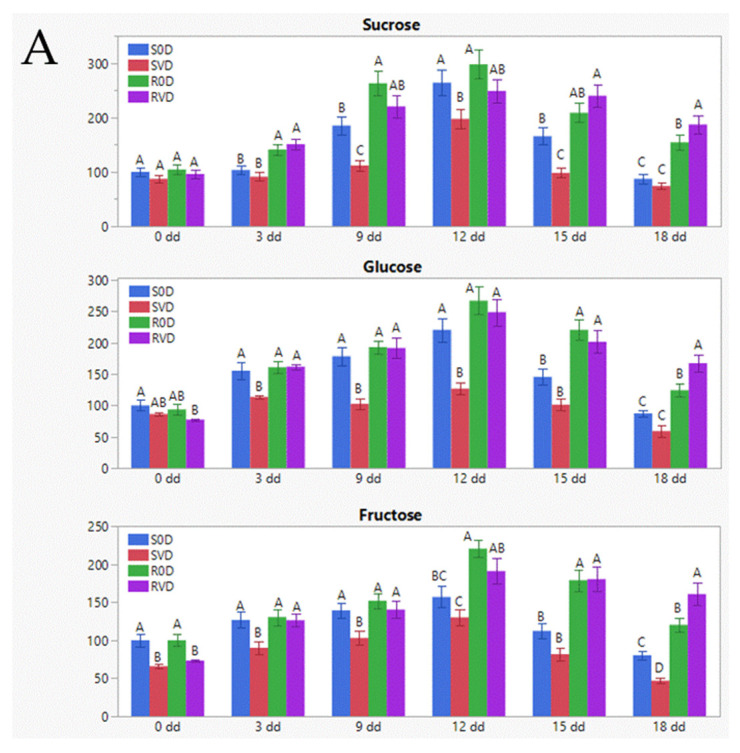
Stabilization of osmoprotectants patterns (sugars, amino acids) in R-GF967 vs. S-967 tomatoes under drought stress. Leaf samples were collected at 0, 3, 6, 9, 12, 15, and 18 dd after the beginning of water withholding from the four groups of experimental tomatoes R0D, RVD, S0D, and SVD. (**A**) Osmo-protective sugar patterns were shown for sucrose, glucose and fructose. The concentration of each sugar in S0D leaf the first day of sampling (0 dd) was considered as 100; the concentration of soluble sugar was calculated relative to this value. Bars represent the standard errors of the relative sugars’ levels from three independent biological repeats. Different lowercase letters (a–d) above the bars denote significant differences (*p* < 0.05). (**B**) Osmo-protective amino acids patterns were shown for proline, HO-proline, valine, tryptophan, γ-aminobutyric acid, asparagine, glutamine, and methionine. The concentration of each amino acid in S0D leaf the first day of sampling (0 dd) was considered as 100; the concentration of each selected amino acid was calculated relative to this value. Bars represent the standard errors of the relative amino acids’ levels from three independent biological repeats. Different capital letters (A–D) above the bars denote significant differences (*p* < 0.05). (**C**) Partial least-squares discriminant analysis (PLSDA) and principle component analysis (PCA) analyses. All the metabolome data for a given genotype at a given time are represented by a colored circle. Red: R0D; green: RVD; light blue: S0D; dark blue: SVD. Contrary to PLSDA, which considers the data supplied for each group, allowing to see the differences in each group, PCA is used for clustering when knowledge is not required. The two graphs show that R0V and RVD cluster together and are different from S0V and SVD.

**Figure 2 cells-10-02875-f002:**
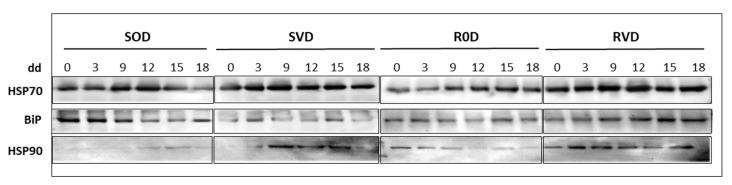
Heat shock proteins (HSPs) patterns shown for HSP70, BiP, and HSP90. Twenty milligrams of dried leaves from three different plants were taken for each sample. Plant tissues were drill-homogenized in 300 µL of SDS-PAGE buffer. Thirty µL of protein extracts were subjected to SDS-PAGE. HSP70, BiP, HSP90 were immuno-detected using specific antibodies. The photographs show the relevant regions of the gel. Three independent biological repeats were used with similar results.

**Figure 3 cells-10-02875-f003:**
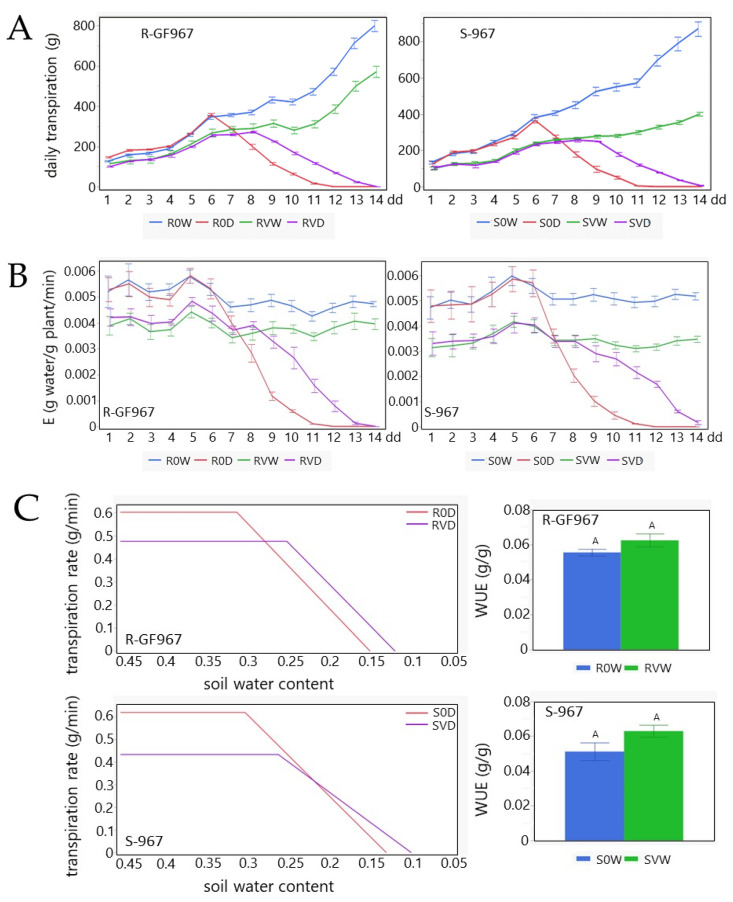
Water balance in S-967 and R-GF967 plants exposed to drought stress depends on virus infection. Using the Plantarray system, physiological parameters were measured over 14 days for uninfected normally irrigated R-GF967 (R0W) and drought-exposed (R0D) tomatoes; the same conditions were analyzed for TYLCV-infected R-GF967 plants (RVW vs. RVD). S-967 tomatoes were coined as S0W-S0D and SVW-SVD. (**A)** Efficiency of daily transpiration expressed as delta of plant mass (g) in morning and evening. Mean ± SE continuous daily whole plant transpiration during the entire experimental period. (**B)** Normalized transpiration (E). Mean ± SE midday transpiration rate normalized to weight (E) during the entire experimental period between 11 am and 1 pm. (**C**) Physiological drought point (Θ_crit_) and water use efficiency (WUE). Θ_crit_ was identified as midday transpiration rate vs. the soil water content (SWC_crit_) of tomatoes grown in water withholding conditions. WUE is identified as the ratio between the gain of daily weight and the daily transpiration. Mean ± SE of WUE calculated automatically by SPAC analytics software; values with same letter are not significant (*p* = 0.05, Student’s *t*-test). (**D**) Calculated plant weight (CPW) determined as the sum of the initial plant weight (calculated before growth in SPAC-Analytics) with the cumulative transpiration multiplied by WUE. Mean ± SE of CPW for the entire experimental period. Values with deferent letters are significant (*p* < 0.05, Student’s *t*-test was carried out separately for normally irrigated and drought on 14 dd).

**Figure 4 cells-10-02875-f004:**
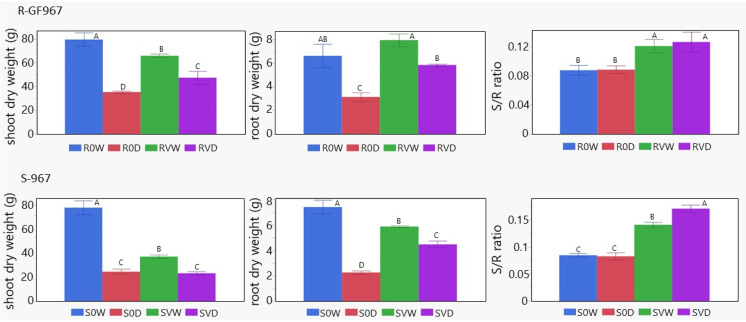
Biomass of TYLCV-infected tomatoes is less affected by drought than of uninfected. Dry weights in grams (g) of shoots and roots of R-GF967 and S-967 tomatoes, harvested at the end of experimental growth, was compared (R0W, R0D, RVW, RVD and S0W, S0D, SVW, SVD). Mean ± SE values with deferent letters are significant (*p* = 0.05, Student’s *t*-test). Root to shoot ratio (R/S) measured for dry weights of tomatoes’ roots and shoots. Mean ± SE of R/S ratio; values with deferent letters are significant (*p* < 0.05, Student’s *t*-test). Three independent biological repeats were used with similar results.

**Figure 5 cells-10-02875-f005:**
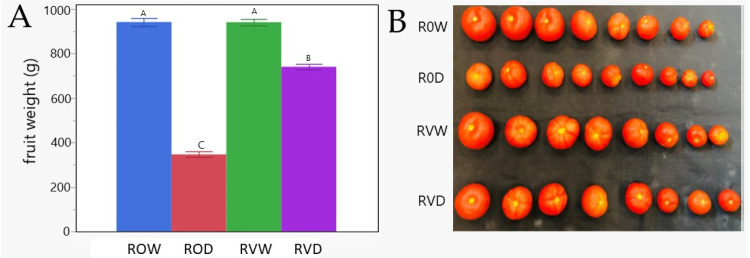
Tomato yield in R-GF967 plants, grown under water deficit, followed by recovery with regular irrigation. After 14 days of growth under drought, R-GF967 plants developed under normal irrigation for another two months. Then all the fruits were collected. (**A**) Average weight of tomato fruits produced by one plant; fruits from five plants randomly collected from each group (HWR, HDR, VWR, and VDR) were gathered, and their fresh weights were measured. Values with deferent letters are significant (*p* = 0.05, Student’s *t*-test). Three independent biological repeats were used with similar results. (**B**) The shape and size of individual fruit from one R0W, R0D, RVW, RVD of R-GF967 plant were compared.

## Data Availability

Data in this study are available from the authors upon request.

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
