# Peer review of "Tomato Yellow Leaf Curl Virus (TYLCV) Promotes Plant Tolerance to Drought"

_cells, 2021, doi:10.3390/cells10112875_

Round 1
Reviewer 1 Report
The article “Tomato Yellow Leaf Curl Virus (TYLCV) Promotes Plant Tolerance to Drought” by Moshik Shteinberg and coworkers is based on previous work by the same group that indicated that TYLCV can dampen heat shock response and inhibit cell death by interacting with heat shock transcription factor HSFA2 and also induces reallocation of carbohydrates and amino acids from shoots to roots. Here they show that tomato plants that are tolerant to TYLCV are more tolerant to draught stress when infected with TYLCV.
The resistant tomato cultivar R-GF967 and the susceptible cultivar S-967 are infected or not with TYLCV and exposed or not to draught stress (14 days iCORE or 18 days in Jordan). Metabolites are analysed for the Jordan plants. Draught phenotype is more severe in healthy than in infected plants, independent of whether the plants are resistant to TYLCV or not. Metabolomic analysis indicates that sugars and amino acids accumulate more in draught-stressed plants, especially after 12 to 15 days. The response seems to be dependent on the tomato genotype with more accumulation measured in the resistant genotype. Infection results in more accumulation of metabolites in resistant plants but not in susceptible plants. Western blot analysis shows no big changes in HSP70 and BiP levels during draught treatment; HSP90 levels are somewhat higher in draught-stressed plants. Viral infection boosts HSP levels moderately, especially in resistant tomato plants. TYLCV infection lowers transpiration in susceptible and resistant plants, but transpiration endures longer in infected plants. Plant mass is a little bit higher in draught-stressed infected plants and lower in watered infected plants, independent of the genotype. Draught-induced loss of shoot and root mass was lower for infected plants, irrespective of the resistance status. However, susceptible infected plants had less mass than healthy susceptible plants. Virus infection of resistant plants did not reduce tomato yields under well-watered condition and yield loss after recovery from draught stress was lower for infected plants than for healthy plants.
Taken together, the authors show that resistant and susceptible tomato plants infected with TYLCV are more resistant to draught stress. However, the resistant infected variety had after draught stress recovery more biomass and fruit than healthy plants. This is an interesting paper showing again that virus infection can also be favourable for plants. It might have direct impact on agricultural practices. The authors argue that the improved draught tolerance is due to enhanced production of osmotic metabolites and to a lesser extent due to accumulation of HSPs.
Author Response
Please, see the attachment

Reviewer 2 Report
In this manuscript, Moshik Shteinberg and colleagues propose that viral infection by TYLCV a ssDNA monopartite Geminivirus protects tomato host plants against extreme drought. The results show that the viral infection produces osmoprotectants (sugars and peptides) that correlate with longer survival during water stress without the consequences after normal irrigation. The experiments are straightforward and appropriate controls have been included. The findings are novel and of great importance to the field of Plant Virology but more insights on which part of the virus is need it before thinking in using this as an application for biotechnology as the authors propose. Other than that, few minor issues listed below will improve the presentation and quality of the manuscript.
Authors might think in why is a bad idea to use the full non-attenuated virus as “treatment” to try to improve productivity. I would opposed to even suggest an attempt to use a virus that causes the most destructive disease in tomato under unknown conditions, only strict biosafety requirements that prove that no arthropod-transmitted virus can escape should allow this kind of experiments. Please specify materials of insect proof cages, containment and biosafety capacities, and how the authors control the single infection of TYLCV by a whitefly virus-fee raring colony.
They have a lot or acronyms that make the reading slow, please consider a better way to present: S-967 R-GF967,HDR, VDR, HDS, VDS, etc.
Table I. It is difficult to read. Needs some reformat and need to be sent to supplemental material. It uses a lot of space and does not provide any new information additional to the text. Presentation of the data needs to improve.
All graphics display days on x-axis, level of metabolite in y-axis and different color for treatment. For your reader this emphasizes they max-min per day, not per treatment. Because the conclusions are made on your treatments and not in your time points these needs to be corrected. Please organize your graphs per treatment grouping the days. This will also help your statistical comparisons.
The p-values should be > or < in some of your figures is = to the actual p-value. Please review https://www.nature.com/articles/nmeth.2698 and note “the P value reported by tests is a probabilistic significance, not a biological one” take this into account to carefully make your conclusions. Also be consistent, if you are defining your values in small letter keep it like that in the graphics too. Again it would improve if you group per treatment instead of per day. Same information just easier to read. Define all your acronyms in the captions (i.e. PCA)
I do not understand your Principal Component Analysis. Which one are your components? Where is the P-value and how was calculated? How fold change was estimated? Please expand.
Figure 2. Please provide a loading control. It can be a total protein load like Ponceau stained or a protein loading control ie proteins that exhibit high-level, constitutive expression in infected-tomato. Loading controls are absolutely essential to ensure the reliability of your data when comparing expression of a protein in different samples.
How do the authors control for whitefly infestation effect. In my opinion a virus-free whitefly treatment is need it.
A simple graphical model to summarize the findings would be helpful, although not necessary
Author Response
Please, see the attachment

Round 2
Reviewer 2 Report
Authors use at their discretion some of the suggestions to improve the manuscript. This version does not address all the comments but greatly improved the previous version and I am happy to recommend this new version for its publication in Cells.